chemical engineering

insulating gas, synthesis, bis-(perfluoroisopropyl) ketone, heptafluoroisobutyramide, heptafluoroisobutyronitrile

**Author for correspondence:**
Yunbai Luo
e-mail: ybai@whu.edu.cn

# Novel and efficient synthesis of insulating gas-heptafluoroisobutyronitrile from hexafluoropropylene

## Zhanyang Gao, Min Wang, Shiyao Wang, Yi Wang, Ruichao Peng, Ping Yu and Yunbai Luo

Engineering Research Centre of Organosilicon Compounds and Materials, Ministry of Education, College of Chemistry and Molecular Sciences, Wuhan University, Wuhan 430072, People's Republic of China

YL, 0000-0002-2833-9802

A novel and efficient preparation route of insulating gas-heptafluoroisobutyronitrile was developed. The synthetic route involved halogen-exchange fluorination, decomposition of bis-(perfluoroisopropyl) ketone and dehydration reaction. Overall, the desired compound was produced without the use of extremely toxic and expensive substances. Structures of the as-obtained products were determined by $^{19}$F NMR, $^{13}$C NMR, IR and GC-MS. The effects of several variables on reaction yield including nature of potassium fluoride, the catalyst dosage, temperature, solvent and molar ratio of raw materials were all investigated. The results indicated that the total yield of heptafluoroisobutyronitrile could reach 42% from original materials under optimal conditions and the mechanism of the reaction was proposed.

## 1. Introduction

$SF_6$ is traditionally employed as an ideal insulating gas in gas-insulated switch gear (GIS) and gas-insulated transmission lines (GIL) owing to its superior chemical stability, excellent insulation properties and arc-extinct performance. However, $SF_6$ is a powerful greenhouse gas ranked as one of the six emission limiting gases in the Kyoto protocol. According to the 2013 Intergovernmental Panel on Climate Change, the estimated Global Warming Potential of $SF_6$ is about 23 500 times that of $CO_2$ [1–5]. Hence, searching for $SF_6$ alternative gases or gas mixtures has received great attention both in industry and academia. In recent years, insulation properties and arc-extinct performances of some gases have been analysed, such as $CO_2$, [6,7] $N_2$, [8,9] $CF_3I$ [10] and c-$C_4F_8$ [11]. However, little attention

has been paid to their physical and chemical properties, greatly limiting their applications. For instance, $CF_3I$ was determined to have better insulation properties than $SF_6$. However, $CF_3I$ [10] could easily decompose under light conditions with poor chemical stability, leading to its extremely short existence in the atmosphere. In addition, iodine could precipitate on the surface of devices after discharge, which would be harmful to equipment. To solve these issues, a g3 gas, [1] a mixture of heptafluoroisobutyronitrile ($C_3F_7CN$) and $CO_2$, was identified after extensive measurements as a good alternative in terms of gas toxicity, insulation properties and greenhouse effect. Therefore, exploration of efficient and green ways to prepare $C_3F_7CN$ has become a top priority in this field.

Numerous efforts have been devoted to finding convenient ways to produce $C_3F_7CN$ and several synthetic routes have been reported. For instance, Tamura *et al*. [12] used perfluorotri-isopropyl-1,2,3-triazine under irradiation (253.7 nm) at 77 K for 2 h to prepare a mixture of $C_3F_7CN$ and perfluoro-2,5-dimethylhex-3-yne. In Simon's production route, [13] $C_3F_7CN$ was prepared by an unusual material, called perfluoro(tri-isopropyl)-1,2,4-triazine. Nevertheless, these methods require complex reaction conditions and equipment. As a result, none of these technologies has been applied commercially because of low efficiency and high cost. Hence, it is desirable to search for efficient synthetic strategies with available materials to produce $C_3F_7CN$ under mild conditions. In 1971, Robert [14] developed a new way of using cyanogen and hexafluoropropene (HFP) to produce $C_3F_7CN$ with good yield. But the toxicity of cyanogen was too high to satisfy requirements of green chemistry. Among those devoted to these efforts, 3M company [15] proposed a method for synthesis of the $C_3F_7CN$ using methyl heptafluoroisobutyrate and $NH_3$ as raw materials to prepare heptafluoroisobutyramide ($C_3F_7CONH_2$). The conversion of $C_3F_7CONH_2$ to $C_3F_7CN$ was then performed through dehydrating agents. However, the methyl heptafluoroisobutyrate was of considerable expense, which would dramatically increase the production cost.

The general methodology for preparing nitriles is dehydration reactions of amides. Therefore, successful synthesis of $C_3F_7CONH_2$ will be the focus of our research. Based on Coffman's research, [16,17] bis-(perfluoroisopropyl) ketone could be produced by reaction of perfluoroisobutyl fluoride with HFP to yield $C_3F_7CONH_2$ after treatment with $NH_3$. However, the toxicity of perfluoroisobutyl fluoride was too high to be handled safely. In 1977, Ishikawa [18] employed oxalyl dichloride, KF and HFP as starting materials to prepare perfluoro-2,5-dimethyl-3,4-hexanedione. But much to their surprise, bis-(perfluoroisopropyl) ketone was generated with 20% yield.

Based on previous studies, we developed a novel synthesis route of $C_3F_7CN$ (figure 1). Oxalyl dichloride, HFP and KF were used as starting agents with 18-crown-6 to prepare $C_3F_7CN$. The route consisted of a three-step reaction, in which structures of products were confirmed. The reaction process was optimized to improve the yield of $C_3F_7CN$ and a mechanism was proposed. Overall, compared with other synthetic methods, the proposed route was easy to operate with low cost, high yield and little pollution. These features indicated that this method had potential for commercial production of insulating gas.

# 2. Material and methods

## 2.1. Materials and general measurements

All solvents used were anhydrous. Oxalyl dichloride (greater than or equal to 98%), acetonitrile (greater than or equal to 99.8%), pyridine (Py), *N*, *N*-dimethyl formamide (DMF), KF·2H$_2$O and KF (ordinary drying process) were all received from Shanghai Aladdin Bio-Chem Technology Company. 18-crown-6 (greater than or equal to 99.0%) was purchased from Shanghai Macklin Biochemical and HFP (greater than or equal to 99.99%) from Chengdu Keyuan Gas Company. KF (spray drying process) was provided by Wuhan Huiyao Tonghui chemical company. $NH_3$ was from Wuhan Xiangyun Industry and Trade company. Trifluoroacetic anhydride (TFAA) was obtained from Energy Chemical company. All other reagents were used as received without further purification.

IR spectra were recorded on a Thermo FT-IR 5700 IR (KBr) [19]F NMR (376 MHz) and [13]C NMR (126 MHz) were recorded on a Bruker Advance-III NMR spectrometer. GC-MS were carried out on a Varian-450 Gas Chromatograph coupled to a Varian-320 Mass Selective Detector equipped with EI detectors. The Gas Chromatograph was equipped with a 30 m and 0.250 mm, 0.25 mm df and VF-5 column. The barricaded pressure vessels of 316 L alloy was obtained from Shanghai Yanzheng Experimental Instrument Co., Ltd (YZPR-250).

**Figure 1.** Three-step synthesis of $C_3F_7CN$ starting from hexafluoropropylene.

## 2.2. Synthesis of first step product: bis-(perfluoroisopropyl) ketone

KF (17.4 g, 0.3 mol), 18-crown-6 (2.4 g, 9 mmol), $CH_3CN$ (60 mL) and oxalyl dichloride (7.6 g, 0.06 mol) were poured into barricaded pressure vessels of 316 L alloy equipped with a mechanical stirrer. The mixture was heated at 60°C for 2 h, and subsequently sealed. Next, HFP (27.0 g, 0.18 mol) was added to the vessel and the mixture was stirred at 90°C for 15 h. Finally, the lower layer of light green liquid was separated by distillation to obtain 15.50 g bis-(perfluoroisopropyl) ketone (72–73°C). The yield was determined as 71% and structures were confirmed by $^{19}F$ NMR, $^{13}C$ NMR and GC-MS.

$^{19}F$ NMR ($CDCl_3$, 376 MHz) $\delta$ -73.14 (d,12F), $\delta$ -188.58 (m,1F).

$^{13}C$ NMR ($CDCl_3$, 126 MHz) $\delta$ 182.00 (t, $J = 21.9$ Hz), 121.04–115.36(m), 92.4(dt, $J = 192.4$, 28.0 Hz).

GC/EI/MS (70 eV) retention time, $m/z$: 1.433 min, 69 ($CF_3^+$), 99.8 ($CF_3CF^+$), 118.8 ($CF_3CF_2^+$), 168.9 (($CF_3)_2CF^+$), 197 (($CF_3)_2CFCO^+$).

## 2.3. Synthesis of second step product: $C_3F_7CONH_2$

$CH_3OH$ (100.0 ml) and $NH_3$ (0.42 mol) were added to bis-(perfluoroisopropyl) ketone (50.0 g, 0.14 mol) at 0°C. After stirring for 6 h, the solvent was removed in vacuum and the residue was crystallized from chloroform (100.0 ml). Colourless crystals of $C_3F_7CONH_2$ were obtained. The yield was determined as 80% while structures were confirmed by $^{19}F$ NMR, $^{13}C$ NMR and GC-MS.

$^{19}F$ NMR ($CDCl_3$, 376 MHz) $\delta$ -74.48 (d, 6F), $\delta$ -180.04∼-180.16 (m,1F).

$^{13}C$ NMR ($CDCl_3$, 126 MHz) $\delta$ 159.64 (d, $J = 17.1$ Hz), 121.49–115.78 (m), 88.44(dt, $J = 182.7$ Hz, 26.8 Hz).

GC/EI/MS (70 eV) retention time $m/z$: 4.115 min, 43.8 ($CONH_2^+$), 69 ($CF_3^+$), 99.9 ($CF_3CF^+$), 169.0 ($M-CONH_2$).

## 2.4. Synthesis of final step product: $C_3F_7CN$

DMF (120.0 ml) and pyridine (62.0 ml, 0.77 mol) were added to a 500 ml three-necked flask at $-5$°C and stirred for 0.5 h. Afterwards, $C_3F_7CONH_2$ (40.0 g, 0.19 mol) was added at the same temperature and continuously stirred for another 0.5 h. The three-necked flask was equipped with constant pressure dropping funnel and dry-ice cold trap. TFAA (53.6 ml, 0.38 mol) was added to the reactants. The reaction temperature was set to 0°C. After stirring for 6 h, the reaction system was heated to 25°C for another 1.5 h. Ultimately, 26.95 g $C_3F_7CN$ was obtained with a yield of 74%. The structures were confirmed by $^{19}F$ NMR, GC-MS and IR.

$^{19}F$ NMR ($CDCl_3$ 376 MHz) $\delta$ -75.35 (d, 6F), $\delta$ -176.59∼-176.76 (m,1F).

GC/EI/MS (70 eV) retention time, $m/z$: 1.447 min, 49.8 ($CF_2^+$), 69 ($CF_3^+$), 75.9 ($CF_2CN^+$), 99.9 ($CF_3CF^+$), 106.9 ($M-(CF_3)F$), 175.9 ($M-F$).

IR: $vC\equiv N = 2272.43$ $cm^{-1}$.

# 3. Results and discussion

## 3.1. Effects of nature of KF on bis-(perfluoroisopropyl) ketone yield

The reaction between **1** and **2** was identified as a nucleophilic fluorination reaction. KF, CsF and NaF were often used as fluorination reagents which had great influence on the reaction conversion. In view of the activity and price, KF was selected as the fluorination reagent. Three types of KF were employed for research: **2a**, **2b** and **2c**. As shown in table 1, **2c** was used as the fluorination reagent achieving the highest yield, followed by **2b**, while the yield of **2a** was 0. According to previous reports [19,20] and experiments, it can be seen that the activity of **2c** was the highest due to its small particle sizes and large specific surface areas, which could be well integrated within 18-crown-6. The

**Table 1.** Effect of nature of 2 on the yield of 3[a].

| entry | 2 | pre-process | | yield (%)[b] |
|---|---|---|---|---|
| 1 | KF·2H$_2$O | drying[e] | **2a** | 0 |
| 2 | KF (ordinary drying process)[c] | drying[e] | **2b** | 30 |
| 3 | KF (spray drying process)[d] | drying[e] | **2c** | 71 |
| 4 | KF (spray drying process)[d] | no drying[f] | **2d** | 55 |
| 5 | KF (spray drying process)[d] | no drying[g] | **2e** | 45 |

[a]Reaction conditions: **1** (0.06 mol), **2** (0.3 mol), HFP (0.18 mol), PTC (9 mmol) in CH$_3$CN (60.0 ml) at 90°C for 15 h.
[b]Isolated yield.
[c]Dried by ordinary process with large particle size (less than 48 μm).
[d]Dried by spray process with small particle size (1.0–15 μm).
[e]KF reagents were dried in muffle furnace at 280°C for 2 h prior to use.
[f]KF reagents were exposed to air for 0.5 h prior to use.
[g]KF reagents were exposed to air for 1 h prior to use.

activity of **2a** was low for the reason that its particle sizes were too large to bond with 18-crown-6. In addition, the effect of water content of KF on yield was also investigated. As illustrated in table 1, the yield fell to less than 50% with water content of KF increasing from 2c to 2e, which was attributed to hydrolysis of oxalyl dichloride. Hence, **2c** was identified as the optimal fluorination reagent.

## 3.2. Effects of reaction conditions on yield of bis-(perfluoroisopropyl) ketone

Reactants were transferred from one phase to another via phase transfer catalyst (PTC) to accelerate reaction rate and improve reactants conversions [21–24]. In solid–liquid systems, the most common used PTC was crown ether. The 18-crown-6 could complex selectively with K$^+$, and F$^-$ was introduced into the organic phase to attack the substrate. The molar concentration of PTC would affect the conversion of **2c** during this process. Table 2 (entries 1–5) shows the effect of catalyst dosage on conversion of **2c**. The yield of **3** was found to increase from 46 to 60% as the catalyst dosage increased from 1 to 3%. The reason was that the concentration of F$^-$ was improved rapidly with increase in the catalyst dosage. However, the yield of **3** stayed stable at around 60% with further addition of the catalyst dosage, indicating that 3 mol% of PTC was enough. Therefore, the optimal molar ratio of PTC to **2c** was 3 mol% (entry 3).

The reaction temperature would also affect the yield of **3** and reaction rate [25,26]. Table 2 (entries 5–10) provided the relationship between reaction yields and temperatures. As shown in table 2, the yield of **3** gradually improved with temperature from 70°C to 90°C. However, the yield was falling rapidly to 35% when the temperature raised to 120°C, attributing to production of more by-products. Thus, the optimal reaction temperature was identified as 90°C (entry 7). It was worth mentioning that this reaction could be scaled up to 150 g in 2.5 l reaction vessel with high yield, and **3** could easily be separated by distillation.

## 3.3. Effects of different reaction conditions on yield of heptafluoroisobutyramide

In Coffman's process, [17] NH$_3$ was directly introduced into **3** without solvent. However, the solvent could not only affect the reaction rate but would also contribute to changing the chemical reaction path, playing a significant role in the reaction [27]. Methanol was selected for this study in view of the melting point of the product and the effects of methanol on yield of **4** were investigated. Table 3 (entries 1,2) showed that yield of **4** significantly improved in methanol solvent under similar conditions. The proton transfer process in solution was promoted by methanol. The solubility of ammonia also increased with better controlled reaction temperature. Many organic processes were reversible and yield would be greatly influenced by concentration of reactants. Therefore, effects of

**Table 2.** Optimized conditions for synthesis of **3**[a].

| entry | PTC (mol%[b]) | $T$[c] (°C) | yield (%)[d] |
|---|---|---|---|
| 1 | 1 | 70 | 46 |
| 2 | 2 | 70 | 55 |
| 3 | 3 | 70 | 60 |
| 4 | 4 | 70 | 59 |
| 5 | 5 | 70 | 60 |
| 6 | 3 | 80 | 64 |
| 7 | 3 | 90 | 71 |
| 8 | 3 | 100 | 60 |
| 9 | 3 | 110 | 51 |
| 10 | 3 | 120 | 35 |

[a]Reaction conditions: **1** (0.06 mol), **2c** (0.3 mol), HFP (0.18 mol) and PTC in $CH_3CN$ (60.0 ml) for 15 h.
[b]The molar ratio of PTC to **2c**.
[c]Reaction temperature.
[d]Isolated yield.

**Table 3.** Effects of different conditions on yield of **4**[a].

| entry | solvent | $NH_3$/**3** | yield (%)[b] |
|---|---|---|---|
| 1 | none | 1 | 34 |
| 2 | $CH_3OH$ | 1 | 59 |
| 3 | $CH_3OH$ | 2 | 68 |
| 4 | $CH_3OH$ | 3 | 80 |
| 5 | $CH_3OH$ | 4 | 80 |
| 6 | $CH_3OH$ | 5 | 80 |

[a]Reaction conditions: **3** (0.14 mol), $NH_3$ in $CH_3OH$ (100 ml) at 0°C for 6 h.
[b]Isolated yield.

$NH_3$/**3** molar ratio from 1 : 1 to 5 : 1 were studied and yields of **4** were displayed in table 3 (entries 2–6). At molar ratios of $NH_3$/**3** from 1 : 1 to 3 : 1, the yield of **4** increased from 59 to 80%. The conversion ratio of **3** remained unchanged with further increase in molar ratio of $NH_3$/**3**. The results of GC indicated that **3** could be completely consumed after stirring for 6 h when the molar ratio of $NH_3$/**3** was 3 : 1. In sum, the optimal molar ratio of $NH_3$/**3** was identified as 3 : 1 (entry 3).

## 3.4. Effect of molar ratio of reactants on the yield of heptafluoroisobutyronitrile

Nitriles are intermediates in organic synthesis, and play important roles in industrial production. The conversion of amides to nitriles could be performed through the use of traditional dehydrating

**Figure 2.** Mechanistic pathway for synthesis of ((a) 3, (b) 4 and (c) 6) heptafluoroisobutyronitrile from hexafluoropropylene.

**Table 4.** Effects of 5/4 molar ratio on yield of 6[a].

| entry | 4 : 5:6 | yield (%)[b] |
|---|---|---|
| 1 | 1 : 1:2 | 46 |
| 2 | 1 : 2:4 | 74 |
| 3 | 1 : 3:6 | 75 |

[a]Reaction conditions: 4, 5 and 6 in DMF (120 ml) at 0°C for 6 h.
[b]Isolated yield.

reagents, such as TFAA, phosphoric anhydride and sulfonyl chloride [28,29]. Here, 4 and 5 were stirred together in presence of Py (amount of Py was twice molar usage of 5) in DMF at 0°C. The conversion of amide to nitrile was monitored by GC. The effect molar ratio of 4/5 was studied and the results were reported in table 4. The optimized molar ratio of 4/5 was identified as 1 : 2 with yield of 74.0%. The yield of 6 did not rise with further increase in molar ratio of 4/5. Some by-products were found by GC-MS. The possible reason may be that occurrence of side reactions limited the increase in yield.

Hence, heptafluoroisobutyronitrile was successfully synthesized via a three-step reaction using oxalyl dichloride, hexafluoropropylene and potassium fluorine as raw materials. The total yield of heptafluoroisobutyronitrile was 42% under the optimal conditions.

## 3.5. Proposed reaction mechanistic

Based on existing literature, [16–17,28–32] and our experimental results, a reaction mechanism was proposed and depicted in figure 2 (**1** to **6**). Oxalyl difluoride was generated by oxalyl dichloride reaction with KF in the presence of 18-crown-6. The formation of perfluoro-2,5-dimethyl-3,4-hexanedione (by-product) also confirmed that the reaction yielded oxalyl difluoride. The produced oxalyl difluoride subsequently decomposed in presence of KF to produce carbonyl fluoride [30,31]. The concentration of $F^-$ greatly increased with $K^+$ continuing to react with PTC. $F^-$ was then added to HFP to form an intermediate that reacted with carbonyl fluoride. Afterwards, **3** was formed by adding perfluoroisobutyl fluoride to HFP [16,17]. **3** was so hindered that its reaction with ammonia cleaved into **4** (figure 2b) [32]. Subsequently, the conversion of amide to nitrile was induced by TFAA-Py systems shown in figure 2c [29].

## 4. Conclusion

In summary, a novel mild and low-cost synthesis route of heptafluoroisobutyronitrile with high yield was developed. Oxalyl dichloride and anhydrous KF could react in presence of 18-crown-6 in acetonitrile to produce bis-(perfluoroisopropyl) ketone without using perfluoroisobutyl fluoride and carbonyl fluoride. Afterwards, bis-(perfluoroisopropyl) ketone was transformed into heptafluoroisobutyramide. Finally, the conversion of amide to heptafluoroisobutyronitrile was ensured by TFAA-Py systems. The overall yield from oxalyl dichloride to heptafluoroisobutyronitrile reached 42% under optimal conditions. Compared with pre-existing methods, production of heptafluoroisobutyronitrile using the proposed method was more efficient and economical.

Data accessibility. The datasets supporting this article have been uploaded as part of the electronic supplementary material.

Authors' contributions. Y.B.L. designed the method; Z.Y.G. synthesized the heptafluoroisobutyramide, heptafluoroisobutyronitrile and wrote the manuscript; M.W. and S.Y.W. synthesized the bis-(perfluoroisopropyl) ketone; Y.W., R.C.P. and P.Y. supported the $^{19}F$ NMR, $^{13}C$ NMR and FT-IR measurements. All authors have given approval the final version of the paper.

Competing interests. We have no competing interests.

Funding. This work was financially supported by the National Key R&D Program of China (2017YFB0902500) and State Grid Science & Technology Project (The Key Technology of Environment-Friendly Gas-Insulated Transmission Line).

Acknowledgements. We thank the Wuhan University Testing Centre for technical assistance.

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
