## [Reviewer comments · Royal Society Open Science]

Review History

RSOS-181751.R0 (Original submission)

Review form: Reviewer 1

Is the manuscript scientifically sound in its present form?

Yes

Are the interpretations and conclusions justified by the results?

Yes

Is the language acceptable?

No

Is it clear how to access all supporting data?

Yes

Do you have any ethical concerns with this paper?

No

Have you any concerns about statistical analyses in this paper?

No

Recommendation?

Accept with minor revision (please list in comments)

Comments to the Author(s)

This manuscript by Luo and coworkers described a three-step synthetic method for the preparation of heptafluoroisobutyronitrile with total 42% chemical yield, by the use of easily available and less toxic hexafluoropropylene, oxalyl chloride, KF and 8-crown-6, which afforded a new and efficient access to such insulating gas, heptafluoroisobutyronitrile. The compounds have been carefully characterized, and the reasonable mechanism has been provided. This method is useful and may be applied in the industry. Thus, this review recommend acceptance of this manuscript for publication on R. Soc. open sci., but after revisions listed below:

1. The manuscript need careful reading, there were many typos and format errors.
2. The schemes in the manuscript: 1) the "+" in all the schemes needed redrawing; 2) Column 3 of Table 3, "NH₃/3 (mol)" makes no sense, please indicate clearly. Maybe "NH₃ (equiv)" is better;
- 3) Figure 2c, the first step is not correct. TFAA should be placed on the arrow.
3. The data list of ¹³C NMR were not correct. The coupling constant should be provided for CF₃, CF, instead of listing all the peaks.
4. The author only mentioned the size of KF particle, whether the water content of KF showed some effect on the yield?

Review form: Reviewer 2

Is the manuscript scientifically sound in its present form?

No

Are the interpretations and conclusions justified by the results?

No

Is the language acceptable?

No

Is it clear how to access all supporting data?

Yes

Do you have any ethical concerns with this paper?

No

Have you any concerns about statistical analyses in this paper?

No

Recommendation?

Major revision is needed (please make suggestions in comments)

Comments to the Author(s)

See attachment (Appendix A).

Decision letter (RSOS-181751.R0)

02-Jan-2019

Dear Professor Luo:

Title: Novel Efficient Synthesis Route of Insulating Gas- Heptafluoroisobutyronitrile from Hexafluoropropylene
Manuscript ID: RSOS-181751

The editor assigned to your manuscript has now received comments from reviewers. We would like you to revise your paper in accordance with the referee and Subject Editor suggestions which can be found below (not including confidential reports to the Editor). Please note this decision does not guarantee eventual acceptance.

Please submit your revised paper before 25-Jan-2019. Please note that the revision deadline will expire at 00.00am on this date. If we do not hear from you within this time then it will be assumed that the paper has been withdrawn. In exceptional circumstances, extensions may be possible if agreed with the Editorial Office in advance. We do not allow multiple rounds of revision so we urge you to make every effort to fully address all of the comments at this stage. If deemed necessary by the Editors, your manuscript will be sent back to one or more of the original reviewers for assessment. If the original reviewers are not available we may invite new reviewers.

On behalf of the Subject Editor Professor Anthony Stace and the Associate Editor Professor John Moses.

RSC Associate Editor:
Comments to the Author:
(There are no comments.)

RSC Subject Editor:
Comments to the Author:
(There are no comments.)

Reviewers' Comments to Author:
Reviewer: 1

Comments to the Author(s)

This manuscript by Luo and coworkers described a three-step synthetic method for the preparation of heptafluoroisobutyronitrile with total 42% chemical yield, by the use of easily available and less toxic hexafluoropropylene, oxalyl chloride, KF and 8-crown-6, which afforded a new and efficient access to such insulating gas, heptafluoroisobutyronitrile. The compounds have been carefully characterized, and the reasonable mechanism has been provided. This method is useful and may be applied in the industry. Thus, this review recommend acceptance of this manuscript for publication on R. Soc. open sci., but after revisions listed below:

1. The manuscript need careful reading, there were many typos and format errors.
2. The schemes in the manuscript: 1) the "+" in all the schemes needed redrawing; 2) Column 3 of Table 3, "NH₃/3 (mol)" makes no sense, please indicate clearly. Maybe "NH₃ (equiv)" is better;
- 3) Figure 2c, the first step is not correct. TFAA should be placed on the arrow.
3. The data list of ¹³C NMR were not correct. The coupling constant should be provided for CF₃, CF, instead of listing all the peaks.
4. The author only mentioned the size of KF particle, whether the water content of KF showed some effect on the yield?

Reviewer: 2

Comments to the Author(s)
See attachment

Author's Response to Decision Letter for (RSOS-181751.R0)

See Appendix B.

RSOS-181751.R1 (Revision)

Review form: Reviewer 1

Is the manuscript scientifically sound in its present form?

Yes

Are the interpretations and conclusions justified by the results?

Yes

Is the language acceptable?

Yes

Is it clear how to access all supporting data?

Yes

Do you have any ethical concerns with this paper?

No

Have you any concerns about statistical analyses in this paper?

No

Recommendation?

Accept as is

Comments to the Author(s)

The authors have addressed the comments and revised the manuscript accordingly. Now, I recommend acceptance of this paper for publication.

Decision letter (RSOS-181751.R1)

15-Feb-2019

Dear Professor Luo:

Title: Novel and Efficient Synthesis of Insulating Gas- Heptafluoroisobutyronitrile from Hexafluoropropylene
Manuscript ID: RSOS-181751.R1

It is a pleasure to accept your manuscript in its current form for publication in Royal Society Open Science. The chemistry content of Royal Society Open Science is published in collaboration with the Royal Society of Chemistry.

On behalf of the Subject Editor Professor Anthony Stace and the Associate Editor Professor John Moses.

RSC Associate Editor:
Comments to the Author:
(There are no comments.)

RSC Subject Editor:
Comments to the Author:
(There are no comments.)

Reviewer(s)' Comments to Author:
Reviewer: 1

Comments to the Author(s)
The authors have addressed the comments and revised the manuscript accordingly. Now, I recommend acceptance of this paper for publication.

Appendix A

Although I am not able to judge if the manuscript entitled „ Novel Efficient Synthesis Route of Insulating Gas- Heptafluoroisobutyronitrile from Hexafluoropropylene”

is written in sufficiently correct English, I am sure that it requires linguistic corrections. Selected sentences requiring adjustment are listed in the attachment. In addition, the manuscript can be shortened by removing a lot of obvious information. Basically, it can be defined as text containing too much unnecessary information. Therefore, in my opinion, this manuscript should be sent to the evaluation only after the rewriting

In recent years, various known gases, such as CO₂,^{6,7} N₂,^{8,9} CF₃I¹⁰ and c-C₄F₈¹¹ have been extensively been tested as insulation gases because of their insulation properties and arc-extinct performances

The method consisted of preparing bis-(perfluoroisopropyl) ketone, then was transformed into heptafluoroisobutyramide.

P3 line 42-43-- Whereas the methyl heptafluoroisobutyrate was too expensive, this would dramatically increase production cost.

P3 line 45 The conversion of amide to nitrile is well known to follow a general step

P3 line 50-57- In 1977, Ishikawa¹⁸ employed oxalyl dichloride, KF and HFP as starting materials to prepare perfluoro-2,5-dimethyl-3,4- hexanedione. The production of bis-(perfluoroisopropyl) ketone was generated accidentally during the experiment. Though the bis-(perfluoroisopropyl) ketone yield was only 20%, it helped us to build our research strategy.

P3 line 60-51- The route consisted of three-steps, in which the structures of were confirmed each of them.

P4 line 2-3- Overall, compared to other synthetic methods, the proposed route was ready availability of reactant, simple synthesis, low cost, high yield with little pollution to environment.

3.3. Synthesis of second step product - C₃F₇CONH₂

CH₃OH (100.0 mL) and NH₃ (0.42 mol) were added to bis-(perfluoroisopropyl) ketone (50.0 g, 0.14 mol) at 0 °C. After stirring for 6 h, the solvent was removed in vacuum and the residue was dissolved in chloroform (100.0 mL) followed by, recrystallization to yield, colourless transparent crystals of C₃F₇CONH₂.

P5 line 48-49- The 18-crown-6 as a PTC could react with K⁺, where F could be employed to attack the substrate. In this process, molar concentration of PTC would impact the reaction yield.

P5 line 53-60- The reason for this had to do with the concentration of F, which increased rapidly with enhancement of catalyst dosage. However, the yield of **3** did not rise with further increase in the catalyst dosage, indicating that 3 mol% of PTC was enough for conversion of **2c**. Further addition of PTC would increase the production cost **3**. Therefore, the optimal molar ratio of conversion of PTC to **2c** was 3 mol% (entry 3).

P6 line 4-13- The yield of **3** gradually improved with temperature from 70 °C to 90 °C, ascribed to increase in vibration frequency and accelerated thermal motion of molecules. However, the yield decreased as temperature further rose (100~120 °C), attributed to production of more by-products. Thus, the optimal reaction temperature was identified as 90 °C (entry 7). Observably, this reaction could be scaled up to 150 g in 2.5 L reaction vessel with higher yield, and **3** could easily be separated by distillation.

Appendix B

Dear Editors and Reviewers:

Thank you for your letter and for the reviewers' comments concerning our manuscript entitled "Novel Efficient Synthesis Route of Insulating Gas- Heptafluoroisobutyronitrile from Hexafluoropropylene" (ID: RSOS-181751). Those comments are all valuable and very helpful for revising and improving our paper, as well as the important guiding significance to our researches. We have studied comments carefully and have made correction which we hope to meet with approval. The main corrections in the paper and the responses to the reviewer's comments are as following:

Responses to the reviewer's comments:

Reviewer :1

Comment 1: The manuscript needs careful reading, there were many typos and format errors.

Response: We are sorry for typos and format errors. We have carefully corrected these mistakes throughout the manuscript according to your comment.

Comment 2: The schemes in the manuscript: 1) the "+" in all the schemes needed redrawing; 2) Column 3 of Table 3, "NH₃/3 (mol)" makes no sense, please indicate clearly. Maybe "NH₃ (equiv)" is better; 3) Figure 2c, the first step is not correct. TFAA should be placed on the arrow.

Response: 1) Thank you very much to point out the "+" in all

scheme's issues in our manuscript. According to comments from you, we redraw the "+" in all schemes.

2) Thank you for your instructive suggestion. We considered that the "NH₃/3 (mol)" represented "the molar ratio of NH₃/3". However, the irrational use of unit(mol) made the "NH₃/3 (mol)" no sense. Now, we have corrected the error in the new manuscript.

3) Thank you very much. According to your comment, we redraw the Figure 2c and TFAA have been placed on the arrow.

Comment 3: The data list of ¹³C NMR were not correct. The coupling constant should be provided for CF₃, CF, instead of listing all the peaks.

Response: Thank you for your valuable advice. We have corrected it in the new manuscript and also list as follows:

Bis-(perfluoroisopropyl) ketone: ¹³C NMR (CDCl₃, 126 MHz) δ 182.00 (t, J=21.9 Hz), 121.04-115.36(m), 92.4(dt, J=192.4, 28.0 Hz).

C₃F₇CONH₂: ¹³C NMR (CDCl₃, 126 MHz) δ 159.64 (d, J=17.1 Hz), 121.49-115.78 (m), 88.44(dt, J=182.7 Hz, 26.8 Hz).

Comment 4: The author only mentioned the size of KF particle, whether the water content of KF showed some effect on the yield?

Response: Thank you very much. According to your comments, we supplement the experiment that the effect of water content of KF on the yield. The experiment results are listed as follows:

Table 1. Effect of nature of 2 on the yield of 3^a

entry	2	Pre-process		yield (%) ^b
1	KF·2H ₂ O	Drying ^e	2a	0
2	KF (ordinary drying process) ^c	Drying ^e	2b	30
3	KF (spray drying process) ^d	Drying ^e	2c	71
4	KF (spray drying process) ^d	No drying ^f	2d	55
5	KF (spray drying process) ^d	No drying ^g	2e	45

^a Reaction conditions: **1** (0.06 mol), **2** (0.3 mol), HFP (0.18 mol), PTC (9 mmol) in CH₃CN (60.0 mL) at 90 °C for 15 h. ^b Isolated yield. ^c dried by ordinary process with large particle size (< 48 μm). ^d dried by spray process with small particle size (1.0~15 μm). ^e KF reagents were dried in muffle furnace at 280 °C for 2 h prior to use. ^f KF reagents were exposed to air for 0.5 h prior to use. ^g KF reagents were exposed to air for 1 h prior to use.

As illustrated in Table 1, the yield was fell to less than 50% with water content of KF increasing from 2c to 2e, which it was attributed to hydrolysis of oxalyl dichloride.

Thanks again for your comments and suggestions.

Review: 2

Comment 1: Although I am not able to judge if the manuscript entitled “Novel Efficient Synthesis Route of Insulating Gas-Heptafluoroisobutyronitrile from Hexafluoropropylene” is written in sufficiency correct English, I am sure that it requires linguistic corrections. Selected sentences requiring adjustment are listed in the

attachment. In addition, the manuscript can be shortened by removing a lot of obvious information. Basically, it can be defined as text containing too much unnecessary information. Therefore, in my opinion, this manuscript should be sent to the evaluation only after the rewriting.

Response: Thank you for your careful reading our manuscript. We have carefully corrected sentences requiring adjustment and polished the manuscript with a professional assistance in writing. Results of our modifications are as follows:

(1) In recent years, various known gases, such as CO₂, N₂, CF₃I and c-C₄F₈ have been extensively been tested as insulation gases because of their insulation properties and arc-extinct performances.

Correction: In recent years, insulation properties and arc-extinct performances of some gases have been analyzed, such as CO₂, N₂, CF₃I and c-C₄F₈.

(2) The method consisted of preparing bis-(perfluoroisopropyl) ketone, then was transformed into heptafluoroisobutyramide.

Correction: The synthetic route involved halogen-exchange fluorination, decomposition of bis-(perfluoroisopropyl) ketone and dehydration reaction.

(3) P3 line 42-43-- Whereas the methyl heptafluoroisobutyrate was too expensive, this would dramatically increase production cost.

Correction: However, the methyl heptafluoroisobutyrate was of considerable expense, which would dramatically increase the production cost.

(4) P3 line 45 The conversion of amide to nitrile is well known to follow a general step.

Correction: The general methodology to preparation nitriles is dehydration reactions of amides.

(5) P3 line 50-57- In 1977, Ishikawa¹⁸ employed oxalyl dichloride, KF and HFP as starting materials to prepare perfluoro-2,5-dimethyl-3,4-hexanedione. The production of bis-(perfluoroisopropyl) ketone was generated accidentally during the experiment. Though the bis-(perfluoroisopropyl) ketone yield was only 20%, it helped us to build our research strategy.

Correction: In 1977, Ishikawa¹⁸ employed oxalyl dichloride, KF and HFP as starting materials to prepare perfluoro-2,5-dimethyl-3,4-hexanedione. But much to surprise, bis-(perfluoroisopropyl) ketone was generated with 20% yield.

(6) P3 line 60-51- The route consisted of three-steps, in which the structures of were confirmed each of them.

Correction: The route consisted of a three-step, in which structures of products were confirmed.

(7) P4 line 2-3- Overall, compared to other synthetic methods, the proposed route was ready availability of reactant, simple synthesis, low cost, high yield with little pollution to environment.

Correction: Overall, compared with other synthetic methods, the proposed route was easily to operate with low cost, high yield and little pollution.

(8) 3.3. Synthesis of second step product - $C_3F_7CONH_2$

CH_3OH (100.0 mL) and NH_3 (0.42 mol) were added to bis-(perfluoroisopropyl) ketone (50.0 g, 0.14mol) at 0 °C. After stirring for 6 h, the solvent was removed in vacuum and the residue was dissolved

in chloroform (100.0 mL) followed by, recrystallization to yield, colorless transparent crystals of $C_3F_7CONH_2$.

Correction: 3.3. Synthesis of second step product - $C_3F_7CONH_2$

CH_3OH (100.0 mL) and NH_3 (0.42 mol) were added to bis-(perfluoroisopropyl) ketone (50.0 g, 0.14 mol) at 0 °C. After stirring for 6 h, the solvent was removed in vacuum and the residue was crystallized from chloroform (100.0 mL). Colorless crystals of $C_3F_7CONH_2$ was obtained.

(9) P5 line 48-49- The 18-crown-6 as a PTC could react with K^+ , where F^- could be employed to attack the substrate. In this process, molar concentration of PTC would impact the reaction yield.

Correction: The 18-crown-6 could complex selectively with K^+ , and F^- was introduced into organic phase to attack the substrate. The molar concentration of PTC would affect the conversion of 2c during this process.

(10) P5 line 53-60- The reason for this had to do with the concentration of F^- , which increased rapidly with enhancement of catalyst dosage. However, the yield of 3 did not rise with further increase in the catalyst dosage, indicating that 3 mol% of PTC was enough for conversion of 2c. Further addition of PTC would increase the production cost 3. Therefore, the optimal molar ratio of conversion of PTC to 2c was 3 mol% (entry 3).

Correction: The reason was that the concentration of F^- was improved rapidly with increase in the catalyst dosage. However, the yield of 3 stayed stable at around 60% with further addition of the catalyst dosage, indicating that 3 mol% of PTC was enough. Therefore,

the optimal molar ratio of PTC to 2c was 3 mol% (entry 3).

(11) P6 line 4-13- The yield of 3 gradually improved with temperature from 70 °C to 90 °C, ascribed to increase in vibration frequency and accelerated thermal motion of molecules. However, the yield decreased as temperature further rose (100~120 °C), attributed to production of more by-products. Thus, the optimal reaction temperature was identified as 90 °C (entry 7). Observably, this reaction could be scaled up to 150 g in 2.5 L reaction vessel with higher yield, and 3 could easily be separated by distillation.

Correction: As shown in Table 2, the yield of 3 gradually improved with temperature from 70 °C to 90 °C. However, the yield was falling rapidly to 35% when the temperature raised to 120 °C, attributing to production of more by-products. Thus, the optimal reaction temperature was identified as 90 °C (entry 7). It was worth mentioning that this reaction could be scaled up to 150 g in 2.5 L reaction vessel with high yield, and 3 could easily be separated by distillation.

In addition, we have deleted some obvious information to make the manuscript more concise.

Once again, thank you for your comments and suggestions.